# Hyperspectral Panoramic Image Stitching Using Robust Matching and Adaptive Bundle Adjustment

**Yujie Zhang, Xiaoguang Mei, Yong Ma, Xingyu Jiang, Zongyi Peng and Jun Huang ***

Electronic Information School, Wuhan University, Wuhan 430072, China; zhangyujie14159@whu.edu.cn (Y.Z.); meixiaoguang@whu.edu.cn (X.M.); mayong@whu.edu.cn (Y.M.); jiangx.y@whu.edu.cn (X.J.); pengzongyi@whu.edu.cn (Z.P.)
* Correspondence: junhwong@whu.edu.cn

**Abstract:** Remote-sensing developments such as UAVs heighten the need for hyperspectral image stitching techniques that can obtain information on a large area through various parts of the same scene. State-of-the-art approaches often suffer from accumulation errors and high computational costs for large-scale hyperspectral remote-sensing images. In this study, we aim to generate high-precision hyperspectral panoramas with less spatial and spectral distortion. We introduce a new stitching strategy and apply it to hyperspectral images. The stitching framework was built as follows: First, a single band obtained by signal-to-noise ratio estimation was chosen as the reference band. Then, a feature-matching method combining the SuperPoint and LAF algorithms was adopted to strengthen the reliability of feature correspondences. Adaptive bundle adjustment was also designed to eliminate misaligned artifact areas and occasional accumulation errors. Lastly, a spectral correction method using covariance correspondences is proposed to ensure spectral consistency. Extensive feature-matching and image-stitching experiments on several hyperspectral datasets demonstrate the superiority of our approach over the state of the art.

**Keywords:** feature matching; hyperspectral images; image stitching

## 1. Introduction

In recent years, UAV-borne hyperspectral remote sensing (HRS) systems have demonstrated great application potential, such as cover classification, side-scan sonar analysis [1], and vegetation mapping [2]. However, to obtain a higher resolution, the field of view must become limited, so it is not suitable for large-scale scenarios [3]. Therefore, the image-stitching technique for HRS urgently needs to align a series of images into a panoramic image.

Image stitching is the process of combining multiple images with overlapping areas into a large panorama [4]. Most common image-stitching methods require precise overlapping fields between images and the same exposure condition to produce a seamless stitching result [5]. The research on image stitching has experienced a long period of development, and many algorithms have been proposed [6]. Recently, researchers have attempted to stitch RGB images, but there are still few pieces of research on UAV-borne hyperspectral image (HSI) stitching technology. Due to the characteristics of UAV-borne HSIs, their performance and efficiency are still not satisfactory [7]. There are some problems in the application of traditional image-stitching algorithms to hyperspectral images, mainly due to the following reasons.

First, in reality, hyperspectral remote-sensing images have complexity, unlike ordinary RGB images. As remote-sensing images usually contain a whole field, forest, lake, and other scenes, repeated structures, weak texture, and even no-texture areas often appear in images. The feature points detected by traditional methods such as SURF [8] are usually densely distributed with the strong texture of the image. In contrast, only a few or even no feature

points can be detected in the region with weak textures, as shown in Figure 1. Since feature-point pairs completely provide the information calculated by the image transformation model, the uneven distribution of feature points may lead to geometric inconsistency between the panorama and the actual scene. Extracting thousands of feature points for correspondence is often necessary for large-scale or high-resolution hyperspectral images, which significantly burdens existing feature-point extraction and matching methods. In order to simultaneously ensure the quantity and quality of the detected feature points, a new feature-point detection and matching algorithm should be proposed to detect rich and accurate feature points in regions with weak textures.

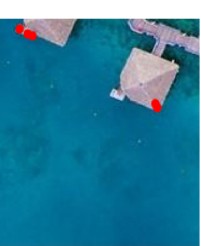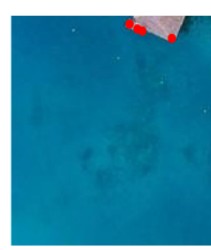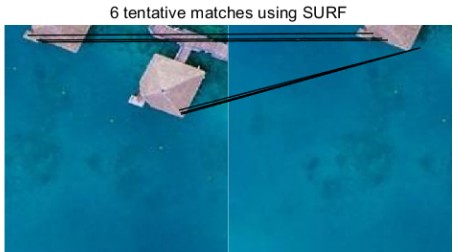

**Figure 1.** In the traditional sample diagram of feature-point detection and matching, feature points can hardly be detected in a region with weak texture, resulting in incorrect matching.

Second, the shooting conditions of UAVs are usually unstable because aircraft roll, yaw, and front and rear jitter are generally inevitable [9]. At the same time, remote-sensing images typically contain some unavoidable local distortion caused by surface fluctuation and changes in imaging viewpoints [10]. So, hyperspectral remote-sensing images often unavoidably contain some parallax that poses higher requirements for feature matching and image alignment accuracy, as shown in Figure 2. If only simple transformations (such as rigid or affine transformations) are used, their matching ability is severely limited.

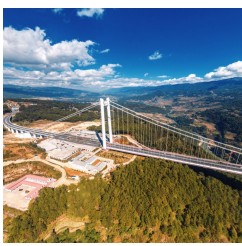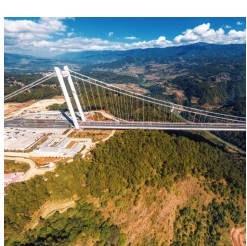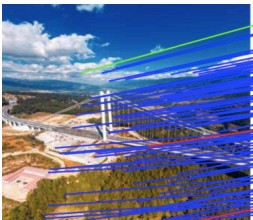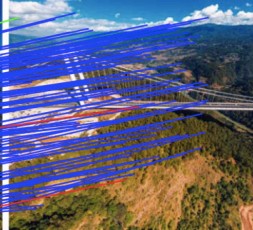

**Figure 2.** Pairs of pictures with parallax and matching sample graphs.

Lastly, hyperspectral image stitching not only considers the alignment of spatial information of each band, but also the spectral consistency of the panorama. The spectrum can reflect the characteristics of the imaging target. This feature can be used to classify and recognize information in images. Therefore, the stitching algorithm for hyperspectral images should also reduce the distortion of spectral data to ensure the integrity of the panorama. Most existing algorithms do not simultaneously consider the spatial and spectral consistency of the stitching results. Existing hyperspectral image-stitching methods have certain limitations. For example, in [11], UAV-based HSIs were stitched using an optimal seamline detection approach. However, this work limited its application to large-scale and multi-image stitching tasks, which presented a severe challenge in solving accumulation errors, and spectral distortion was noticeable.

This article proposes an effective and robust method to obtain a high-precision panorama for large-scale UAV-borne hyperspectral images. We concentrate on both improving alignment accuracy and reducing spectral distortion. The main contributions in this work are as follows:

1.  First, a feature extraction method based on SuperPoint is introduced to simultaneously improve the quantity and quality of feature points. Then, a robust and fast mismatch removal approach, linear adaptive filtering (LAF), is used to establish accurate correspondences that can handle rigid and nonrigid image deformations, and avoid much calculation.

2.  Second, an adaptive bundle adjustment by continually reselecting the reference image was designed to eliminate the accumulation of errors.

3.  Lastly, a covariance-correspondence-based spectral correction algorithm is proposed to ensure the spectral consistency of the panorama.

This article is organized as follows. Section 2 presents the related works. Section 3 proposes an effective and robust method to obtain a high-precision panorama for large-scale UAV-borne hyperspectral images. Then, Section 4 outlines conducted comparative experiments on the algorithms of each part, and our final panorama result of 54 images is presented. Lastly, we summarize the full text in Section 5.

## 2. Related Works

The most used solutions for image stitching are feature-based methods, including feature detection, feature matching, and image alignment [12]. Feature detection is the first step upon which the stitching performance relies heavily. Specifically, some feature extractors are based on spectral and spatial information, such as SS-SIFT [13]. However, large data can require huge computation times for hyperspectral images with hundreds of bands. Since each band of hyperspectral images contains the same spatial information, it is reliable to obtain a single band and extract features from that specific band. The most used feature detection methods, such as SIFT [14] and SURF [8], are invariant to rotation, scale transformation, and brightness changes. However, feature points are clustered and distributed unevenly in overlapping areas by handcrafted approaches, thus impacting the performance of applications. Recently, with the rapid development of deep-learning techniques, some deep descriptors [15,16] have shown superiority over handcrafted ones in different tasks [17].

For UAV-borne hyperspectral images with great spatial resolution, mismatch problems are inevitable, so it is not easy to obtain reliable matching points [18]. Most of the existing mismatch removal approaches use RANSAC [19], but it is unsuitable for nonrigid situations and has limited success. Methods based on local geometry constraints, such as LLT [20], LPM [21], and GLPM [22], are very successful in many situations. However, UAV-borne hyperspectral remote-sensing images usually contain some parallax caused by a change in imaging viewpoint, which greatly affect these feature-matching methods. In order to solve this problem, nonparametric fitting methods such as mTopKRP [23] were proposed. However, when the proportion of outliers is assumed to be too large, it seriously degenerates. Those approaches are usually time-consuming, especially for large-scale or high-resolution images, so the computation time becomes a serious problem.

In feature-based image-stitching methods, establishing a parametric image alignment model is crucial among those steps. The representative work is AutoStitch [24], which uses a single homography to realize the stitching of multiple parallax-free images. However, when the camera translation cannot be ignored or the scene is not located near the plane, the global transformation model does not work well. This is because a global homography cannot consider all the alignments at different depth levels. Some methods using the local transformation model have been proposed to solve the parallax problem. ANAP [25] computes the local transformation models of each image, but the perspective distortions in the nonoverlapping regions still exist. The NISwGSP [26] stitching approach guides the warping transformation models with a grid mesh. ELA [27] uses a regional warp to eliminate parallax errors. However, to better solve the parallax problem, we want to divide the image into smaller subregions to move it close to the plane, so the correspondences on these subregions are very rare. So, while optimizing the alignment model, the quantity and quality of feature points also need to be ensured.

To improve the stitching accuracy, bundle adjustment is often used to optimize a model. In many methods, to avoid accumulated errors, additional sensors are required to directly obtain the camera pose. However, in practice, satellite and drone imagery parameters are not always available [28]. Xing et al. [29] proposed a method based on minimal registration error that extends the Kalman filter to local regions and then globally refines the parameters. However, due to the parallax of the ground scene, this method is not suitable for the image stitching of large areas. Xia et al. [30] first aligned the images via an affine model and then performed model refinement under inverse perspective constraints. However, always fixing the first image as the reference image is unreasonable in large-scale stitching tasks, as it leads to a rapid accumulation of errors.

## 3. Methodology

This section details our proposed algorithm, including feature extraction and matching, adaptive bundle adjustment, spectral correction, and multiband blending. The flowchart of our method is shown in Figure 3.

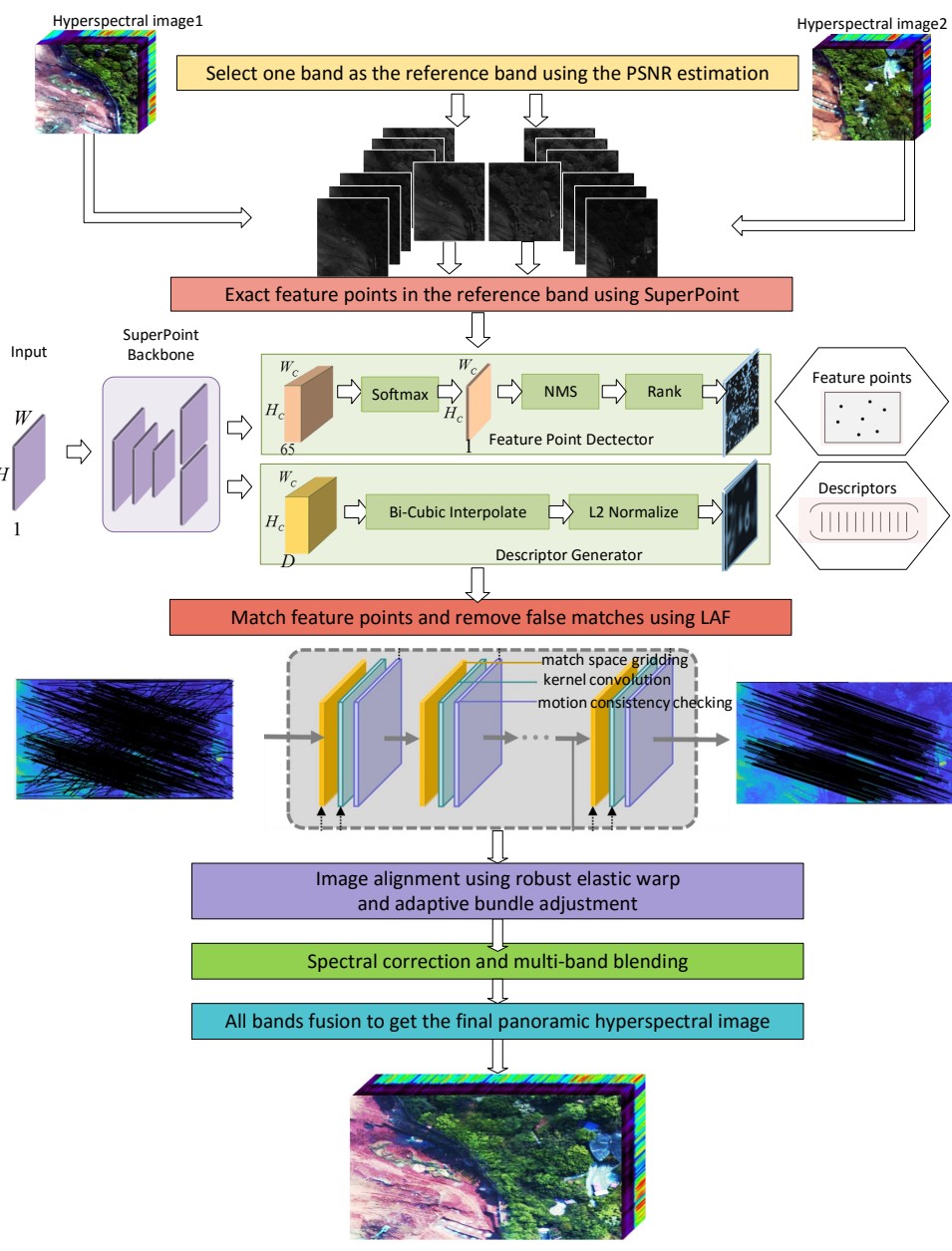

**Figure 3.** Proposed flow of hyperspectral image stitching.

*3.1. Feature Extraction and Matching*

Hyperspectral images contain hundreds of bands, so extracting critical points for all bands takes a lot of time. The spatial information in each band is the same, so selecting a band as the reference can significantly reduce the calculation time. Feature extraction algorithms are sensitive to image noise. To accurately extract image features, feature points are extracted in the band with the slightest noise. This article uses the method based on peak signal-to-noise ratio (PSNR) estimation for band selection.

3.1.1. Self-Supervised Interest-Point Detection and Description

Feature extraction is the first and most crucial step in image stitching because feature points provide the information used in image transformation model calculation. Ideally, feature points are evenly distributed in overlapping areas, and stable features can be extracted in the case of visual angle changes and illumination changes. Feature descriptors are usually the final form of image feature extraction. Good feature descriptors can capture stable and discriminating information of images. In this paper, the SuperPoint network is used to extract robust feature points and feature descriptors, and the feature could achieve good accuracy.

A fully convolutional model, the SuperPoint network [31], was adopted to extract feature points. The main advantages of the SuperPoint network are as follows: First, SuperPoint can detect feature points evenly distributed in the overlapping region, rendering it more suitable for textureless areas in hyperspectral remote-sensing images. Second, instead of patch-based networks, the input to the network is full-sized images.

First, the network maps input image $I \in \mathbb{R}^{H \times W}$ to an intermediate tensor $\mathcal{B} \in \mathbb{R}^{H_c \times W_c \times F}$. The calculation is simultaneously divided into two headers: a 2D interest point detector head and a descriptor head. The 2D detector head computes $\mathcal{X} \in \mathbb{R}^{H_c \times W_c \times 65}$. After channelwise softmax and nonmaximal suppression (NMS), [32], the detector sorts the detected feature points according to the given confidence. It selects $k$ feature points with the highest confidence as the output. To reduce computation and memory, instead of using dense methods, the descriptor learns semidense descriptors $\mathcal{D} \in \mathbb{R}^{H_c \times W_c \times D}$. Then, the bicubic interpolation algorithm is performed to obtain complete descriptors of size $\mathbb{R}^{H_c \times W_c \times D}$, and lastly, L2 normalization is used to obtain unit length descriptions.

Let $I_l$ and $I_r$ be a pair of images to be stitched, $x_{l,r}^k$ and $x_{r,l}^k$ represent feature points. $P_l(x_{l,r}^k)$ denotes the feature descriptor for $x_{l,r}^k$, and $P_r(x_{r,l}^k)$ is the feature descriptor of $x_{r,l}^k$.

3.1.2. Robust Feature Matching via Linear Adaptive Filtering

For large-scale or high-resolution hyperspectral images, extracting thousands of feature points for correspondence is often necessary, which significantly burdens existing feature-point extraction and matching methods. Since remote-sensing images inevitably produce some surface fluctuation and local deformation due to the change of imaging viewpoint, if only rigid transformation is used, the matching of remote-sensing images is severely limited. We adopted the LAF [33] strategy to effectively eliminate mismatches, which is effective for hyperspectral remote-sensing images. This approach can deal with rigid and nonrigid image deformation. At the same time, the grid strategy render the method linear in time and space complexity. Even assuming the set contains thousands of matches, the matching problem can be completed in tens of millimeters. This time is conducive to solving large-scale, real hyperspectral remote-sensing image-stitching tasks.

Inspired by the theory of filtering and denoising, this method uses an a priori geometric consistency to detect outliers after establishing putative feature correspondences by local descriptor features and eliminating them. First, the algorithm divides the putative feature corresponding space into grids. Then, the average motion vector of each cell is calculated. Next, we can obtain the typical motion vector using Gaussian kernel convolution. Lastly, we can calculate the consistency with a threshold detection method to obtain the final inlier set.

According to the principle of image denoising, the distribution and size of noise in the image are irregular and random. Similarly, in feature-point removal, potential true matches tend to be regular and smooth. First, assume that the putative set of matches $S$ is transformed into $S'$:

$$S' = \{(\mathbf{x}_{l,r}^k, \mathbf{m}_k)\}_{k=1}^n, \tag{1}$$

where $\mathbf{m}_k = \mathbf{x}_{r,l}^k - \mathbf{x}_{l,r}^k$ is the motion vector of $(\mathbf{x}_{l,r}^k, \mathbf{x}_{r,l}^k)$, $n$ is the number of putative matches. Then, we equally divide feature points $\chi = \{\mathbf{x}_k\}_{k=1}^n$ into multiple dimensions $n_c \times n_c$. Accordingly, $S'$ can be divided into $n_c \times n_c$ parts with $\chi = \{C_{j,k}\}_{j,k=1}^{n_c}$. Then, a Gaussian kernel distance matrix $\mathbf{K}$ of size $n_k \times n_k$ is defined. Then, we initialize the parameters using the following equation:

$$\begin{cases} n_c = \min(\{\max\{[\sqrt{n_s}], 15\}, 30\}) \\ n_k = \mathrm{odd}(n_c/3), \end{cases} \tag{2}$$

$$\mathbf{K}_{i,j} = \frac{\exp\{-\mathbf{D}_{i,j}\}}{\sum_{i=1}^{n_k} \sum_{j=1}^{n_k} \exp\{-\mathbf{D}_{i,j}\}}, \quad \mathbf{D}_{i,j} = \parallel \mathbf{s}_{i,j} - \mathbf{s}^* \parallel^2, \tag{3}$$

where $n_s$ is the number of the putative match set, $\mathbf{s}_{i,j} = (i,j)^T$ denotes the corresponding position in the convolutional kernel $\mathbf{K}$, and $\mathbf{s}^* = (\lceil n_k/2 \rceil, \lceil n_k/2 \rceil)^T$ denotes the central position. After we transform the putative matching set $S$ into $S'$, we grid the putative corresponding space and calculate the average motion vector of each cell. To remove outliers progressively, the next step is an iteration strategy. Generated matrix $\tilde{\mathbf{M}}$ after Gaussian kernel convolution is calculated with:

$$\tilde{\mathbf{M}} = \frac{(\mathbf{V} \cdot \bar{\mathbf{M}}) \otimes \mathbf{K} - \bar{\mathbf{M}} \cdot \mathbf{K}^*}{\mathbf{V} \otimes \mathbf{K} - B(\mathbf{V}) \cdot \mathbf{K}^* + \varepsilon} \tag{4}$$

where $B(\mathbf{V})$ is the binary form of $\mathbf{V}$, $\bar{\mathbf{M}}$ is the average motion matrix, $\mathbf{K}^*$ is the central element, and $\varepsilon$ is a tiny positive number n in the case that the denominator is 0. $\mathbf{K}$ is a Gaussian kernel distance matrix, and $\mathbf{V}$ is a count matrix with $\mathbf{V}_{j,k} = \mid C_{j,k} \mid$. The convolution gives us a typical motion vector for each cell; then, we define the deviation between $\mathbf{m}_i$ and $\tilde{\mathbf{M}}_{j,k}$. We constrain them with a point value between 0 and 1, and obtain the following formula:

$$d_i = 1 - \exp\{-\frac{\parallel \mathbf{m}_i - \tilde{\mathbf{M}}_{j,k} \parallel^2}{\beta^2}\}, \ \forall i \in C_{j,k}, \tag{5}$$

where $\beta$ is used for determining the width of the interaction range between the two motion vectors, and we empirically set $\beta^2 = 0.08$. Thus, inlier set $R^*$ is roughly obtained.

$$R^* = \{(\mathbf{x}_{l,r}^k, \mathbf{x}_{r,l}^k) : d_i \leq \lambda\}. \tag{6}$$

According to the posterior probability estimation, we can obtain

$$p_i = \begin{cases} 0, & d_i > \lambda \\ 1, & d_i \leq \lambda. \end{cases} \tag{7}$$

Then, $\tilde{\mathbf{e}}_i = \mathbf{m}_i - \tilde{\mathbf{M}}_{j,k}, \forall i \in C_{j,k}$. We can obtain

$$\sigma^2 = \frac{\mathrm{tr}(\mathbf{E}^T \mathbf{P} \mathbf{E})}{2 \cdot \mathrm{tr}(\mathbf{P})} \tag{8}$$

$$\gamma = \frac{\mathrm{tr}(\mathbf{P})}{n} \tag{9}$$

where $\mathbf{P} = \mathrm{diag}(p_1, \cdots, p_n)$ is a diagonal matrix, and $\mathbf{E} = (\tilde{\mathbf{e}}_1, \cdots, \tilde{\mathbf{e}}_n)^T$, $\mathrm{tr}(\cdot)$ is the trace of a matrix. Next, we calculate probability $p_i$ on the basis of the Bayesian rule:

$$p_i = \frac{\gamma e^{-\{\frac{\|\mathbf{m}_i - \tilde{\mathbf{M}}_{j,k}\|^2}{\beta^2}\}}}{\gamma e^{-\{\frac{\|\mathbf{m}_i - \tilde{\mathbf{M}}_{j,k}\|^2}{\beta^2}\}} + \frac{2\pi\sigma^2(1-\gamma)}{a}} \tag{10}$$

Lastly, inner set $R^*$ can be calculated with the following formula using predefined threshold $\tau$, and we empirically set $\tau = 0.8$:

$$R^* = \{(\mathbf{x}_{l,r}^k, \mathbf{x}_{r,l}^k) : p_i > \tau\}, \tag{11}$$

LAF filters outliers step by step using an iterative strategy. In the hyperspectral stitching task, we set the number of iterations to 5 in each iteration. We also set $\lambda = 0.8$, 0.2, 0.1, 0.05, and 0.05 in each iteration. Experiments show that our method is robust to hyperspectral remote-sensing data.

### 3.2. Adaptive Bundle Adjustment

Given a group of matched feature points $\{\mathbf{x}_{l,r}^k, \mathbf{x}_{r,l}^k\}$ in two adjacent images to be stitched, image alignment is based on a transformation model to estimate the mapping relationship. The deformation of image $I_r$ is represented as $\mathbf{G}(x,y) = (g(x,y), h(x,y))^T$, where $g(x,y)$ and $h(x,y)$ are deformations in the $x$ and $y$ directions. We use $\mathbf{p}_l^{i'} = (x_i', y_i')^T$ to denote projection $\mathbf{p}_l^i$ in $I_r$ by Brown and Lowe [24]. For ideal cases without parallax, projection $\mathbf{p}_l^{i'} = \mathbf{p}_r^i$, which means that global transformation $\mathbf{H}$ can estimate a sufficiently accurate alignment. However, projection errors arise for general cases of parallax. The parallax error on $\mathbf{p}_l^{i'}$ is represented by projection bias $\mathbf{G}_i = \mathbf{p}_l^{i'} - \mathbf{p}_r^i = (g_i, h_i)^T$. Assuming that there are $I_1$, $I_2$, ..., $I_n$, the transformation matrices between adjacent images are $\mathbf{H}_{12}$, $\mathbf{H}_{23}$, $\mathbf{H}_{(n-1)n}$. $\mathbf{H}_{ij}$ is the transformation matrix of $I_j$ to $I_i$, and the transformation matrix of the new target image and the reference image can be calculated accordingly: $\mathbf{H}_m = \prod_{i=r}^{n-1} \mathbf{H}_{i(i+1)}$. Although this process is simple, computation-intensive, and time-consuming, due to the multiplicative property of multiplication, errors accumulate very quickly, leading to severe distortions in the final panorama. Regarding parallax and accumulation errors, an adaptive bundle adjustment was designed to provide more accurate and reliable alignment.

First, we used the robust elastic warp [34] to avoid the parallax. The transformation is the combination of the homography and similarity:

$$\mathbf{H}_q = \mu_h \mathbf{H} + \mu_s \mathbf{H}_s, \tag{12}$$

where $\mathbf{H}$ is the homographic transformation, and $\mathbf{H}_s$ is the similarity transformation. $\mu_s$ linearly decrements from 1 to 0 of the source image, with $\mu_h$ from 0 to 1. Target image $I_2$ is transformed as follows:

$$\mathbf{W} = \mathbf{H}_q \mathbf{H}^{-1}. \tag{13}$$

However, for multi-image and large-scale scenes, errors can accumulate very quickly. Considering parallax and accumulation errors, it is not reasonable to use the first image as the reference throughout the process [35]. So, a weighted graph is constructed to update the reference image, which is defined as:

$$A_{ij} = \frac{1}{\log(\text{keyNums} + o)}, \tag{14}$$

where *keyNums* is the number of matching points between $I_i$ and $I_j$, and $o$ is set as a constant, which was set to be 50 in our experiments. We could construct the shortest path cost matrix $\mathcal{A}$ where each element represents the cost of the shortest path between two adjacent images. Therefore, the element with the lowest cumulative cost in $\mathcal{A}$ is treated as the reference image.

After each update of the reference image, the alignment model between the images to be stitched is recalculated in the optimization framework. The transformation set $\mathcal{W}$ of images to be stitched can be solved as follows:

$$E(\mathcal{W}) = \sum_{I_i \in \mathcal{G}} \sum_{k=1}^{keyNums_{i,ref}} \|\mathbf{W}_i \mathbf{x}_{i,ref}^k - \mathbf{W}_{ref} \mathbf{x}_{ref,i}^k\|^2 \tag{15}$$

where $\mathcal{G} = \{I_i\}_{i=1}^m$ is the set of currently aligned images, $keyNums_{i,ref}$ is the total number of matches $\{\mathbf{x}_{i,ref}^k, \mathbf{x}_{ref,i}^k\}$ between $I_i$ and $I_{ref}$.

In order to speed up computation, the optimal solution of nonlinear least-squares Equation (13) can be solved with the sparse Levenberg–Marquardt algorithm [36]. Since the continuous updating of the reference image reduces error accumulation, and the bundle adjustment enables accurate alignment, our algorithm dramatically improved global alignment accuracy.

### 3.3. Spectral Correction and Multiband Blending

After the spatial stitching of single-band images, their spectral information also needs to be matched. We then fused the spectral information of all bands to realize the spectral correction. Each point in the overlapping area was a pair of the correspondence of two adjacent images. In hyperspectral images, each point of a single-band image has a spectral value that is different from its correspondence point. In order to realize the spectral matching of images, this difference must be corrected. In two adjacent images of one band, the wavelength is the same, and the spectral values of correspondence points are different. According to this difference, the new spectral value of each point in the overlapping region can be obtained, which is the spectral correction:

$$Y' = \frac{Y + \left\{ \sum_{x_s \in S}[CN(x_t) - CN(x_s)] \right\}}{keyNumInS}, \tag{16}$$

where $Y'$ represents the spectrum after correction, and $Y$ represents the original one. $x_s$ is the feature point in the reference image, and $x_t$ is the corresponding matching feature point in the target image. $keyNumInS$ is the total number of feature points in the overlapping area, and $CN(x_t)$ and $CN(x_s)$ are the average gray values in the $3 \times 3$ region around feature points $x_s$ and $x_t$. The spectrum of the nonoverlapping area maintains the original one. When the new spectral values are obtained at each point of the overlapping area, the final spectrum after spectral matching is obtained.

Next, we fused multiple images using the multiband fusion method, and calculated weight map $W_{max}^i(\theta, \phi)$ in a spherical coordinate system:

$$W_{max}^i(\theta, \phi) == \begin{cases} 1, & if\ W_{max}^i(\theta, \phi) = \arg\max W_{max}^i(\theta, \phi) \\ 0, & otherwise, \end{cases} \tag{17}$$

$W_{max}^i(\theta, \phi)$ is 1 if the image to be stitched has maximal weight, and 0 where some other image has a higher weight.

Given $N$ input hyperspectral images $I_p$, $p = 1, \cdots, N$ with $L$ bands, and the matching points between images $\{\mathbf{x}_{l,r}^k, \mathbf{x}_{r,l}^k\}$, the workflow is presented in Algorithm 1.

---

**Algorithm 1** Hyperspectral panoramic image stitching using robust matching and adaptive bundle adjustment

---

**Require:** Hyperspectral images $I_p$, $p = 1, ..., N$ each having $L$ bands
**Ensure:** Hyperspectral panorama with high alignment accuracy and low spectral distortion

1: The PSNR of each band is calculated to select the reference band.
2: Extract the feature points by SuperPoint and obtain a putative match set $S$ between two adjacent images set in the reference band.
3: Use kernel convolution to generate a typical motion field using Equations (4) and (5)
4: Obtain inlier set $R^*$ and matching points $\{\mathbf{x}_{l,r}^k, \mathbf{x}_{r,l}^k\}$ by adaptive filtering using Equation (11)
5: **for** $p = 1$ to $N - 1$ **do**
6:    $q = p + 1$
7:    Set $I_1$ as reference image $I_{ref}$.
8:    Update reference image $I_{ref}$ using Equation (14).
9:    Formulate Equation (15) on the basis of matches $\{\mathbf{x}_{q,ref}^k, \mathbf{x}_{ref,q}^k\}$.
10:    Correct the hyperspectral spectral information of all bands using Equation (16).
11:    Align $I_p$ and composite it to the panorama.
12:    Map $I_p$ to $I_{ref}$ using Equation (17).
13: **end for**
14: Stitch the panorama using the same transformation model for the remaining $(L - 1)$ bands.

---

## 4. Experiments and Analysis

In this section, we compare our proposed method with the state of the art using hyperspectral and remote-sensing image datasets. First, the dataset used in the experiment is introduced. Then, our algorithm is compared with existing algorithms, and feature-matching and image-stitching results are analyzed. Lastly, we obtain our panoramic hyperspectral image and analyze the spectrum of the overlapping region.

### 4.1. Datasets

We used five image datasets to evaluate our algorithm in which the HSI dataset was from [34]. This dataset contains 54 UAV-borne hyperspectral images (HSI) with a size of $960 \times 1057$ and spectral range from 400 to 1000 nm, with a total of 176 bands. We tested feature-matching performance on images with different transformations. There are few hyperspectral remote-sensing image datasets labeled with ground truth. In addition, feature points are extracted on a single band. So, we used four remote-sensing datasets from [23] to test the feature extraction and matching performance, comprising 40 pairs of $700 \times 700$ color infrared aerial photographs (CIAP), 25 pairs of $600 \times 337$ unmanned aerial vehicle (UAV) images, 34 pairs of $256 \times 256$ or $800 \times 800$ synthetic aperture radar (SAR) images, and 30 pairs of $1280 \times 1024$ or $1088 \times 1088$ fisheye (FE) images that undergo projective, similarity, projective, rigid, and nonrigid transformations.

### 4.2. Results on Feature Matching

First, we evaluated the performance of some of the most popular handcrafted and deep-learning-based local features on hyperspectral image (HSI) datasets. The results of feature detection and matching are shown in Figure 4. SIFT [14] detected many feature points, but the feature points were concentrated in areas with complex textures. That is to say, there were few feature points and even fewer correspondences in textureless areas. SURF [8] and KAZE [37] provided too few matches to meet the subsequent calculation needs. SuperPoint [31] improved both the quantity and quality of feature points, evenly distributing them in the overlapping region.

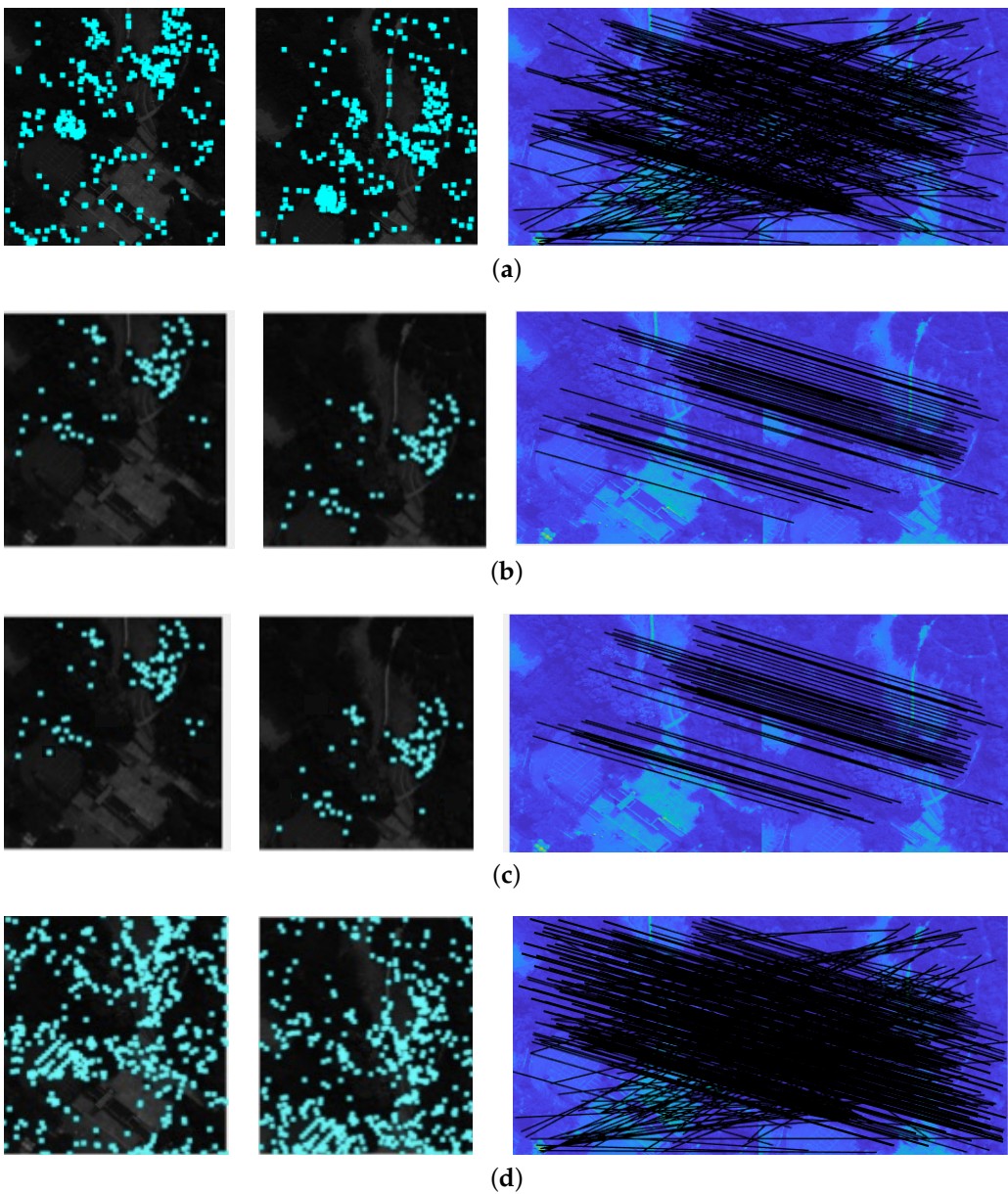

**Figure 4.** Feature point detection and matching of SIFT [14], SURF [8], KAZE [37] and SuperPoint [31]: 1275 future points and 754 tentative matches by SIFT [14], 275 future points and 132 tentative matches by SURF [8], 281 future points and 154 tentative matches by KAZE [8], 2246 future points and 1756 tentative matches by SuperPoint [31]. (**a**) SIFT; (**b**) SURF; (**c**) KAZE; (**d**) SuperPoint.

Then, we tested the performance of our proposed LAF and compared it with that of other representatives, namely, RANSAC [19], VFC [38], LPM [21], and mTopKRP [23], on four datasets with both rigid and nonrigid transformation. The performance values are summarized in Figure 5. *F-score* was determined as the harmonic mean of precision and recall, equal to $2 \times Precision \times Recall / (Precision + Recall)$. Each column in Figure 5 represents the results of a rigid, projected, and nonrigid dataset from left to right. For rigid datasets, all methods achieved high accuracy because there was only a simple rigid transformation between these images. However, for nonrigid datasets, although RANSAC achieved high accuracy, the method in this paper could retain more correct matches, resulting in better *Recall*. mTopKRP also achieved good results, but its running time was too long, and its efficiency was low. The LPM had the shortest running time, but when the putative set involves many outliers, and inliers are distributed dispersedly, the performance (especially *Precision*) begins to sharply worsen. Through the *Fscore*, results show that our

LAF was the best and had obvious advantages. In terms of accuracy and recall rate, our LAF could achieve good results, while the robustness of other methods was poor. Compared with other algorithms, LAF also had high effectiveness.

We conducted a comparative experiment on the HSI dataset to prove the effectiveness of SuperPoint features and the LAF algorithm in our approach. The results are given in Figure 6. The images selected in this paper are challenging for a feature-matching task. The first and third lines had projection distortion, the second line had severe noise, the fourth line had a small overlap area, and the last line had nonrigid distortion. Figure 6 shows that our method successfully identified most of the real matching pairs with only a few errors. These visual results show that our LAF could handle different situations even with large parallax, and is suitable for remote-sensing image-stitching tasks.

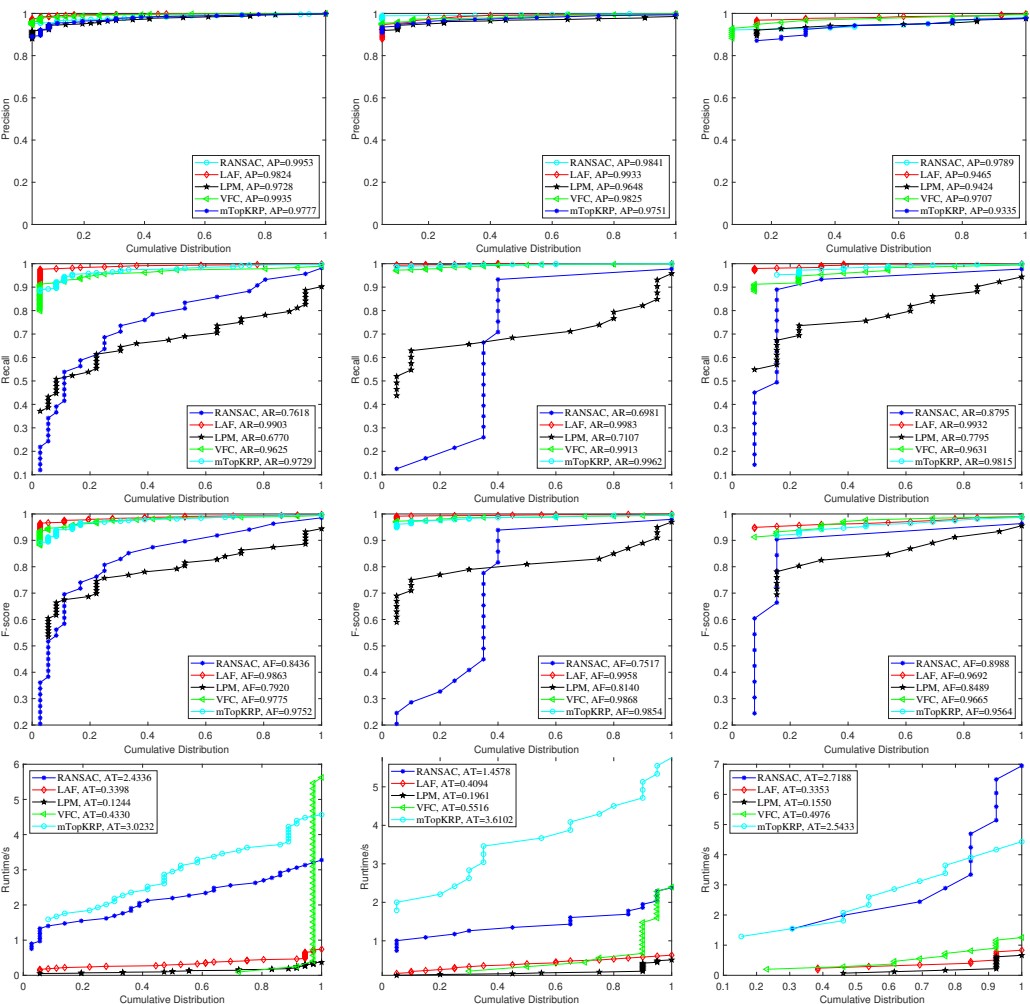

**Figure 5.** Quantitative comparisons of RANSAC [19], VFC [38], mTopKRP [23], LPM [21] and our LAF on on five image sets: (from (**left**) to (**right**)) rigid (SAR, CIAP), projection (UAV) and nonrigid (FE). ((**top**) to (**bottom**)) *Precision*, *Recall*, *F-score*, and *Runtime* with respect to the cumulative distribution. The average *Precision* (AP), average *Recall* (AR), average *F-score* (AF), and average *Runtime* (AT) are reported in the legend.

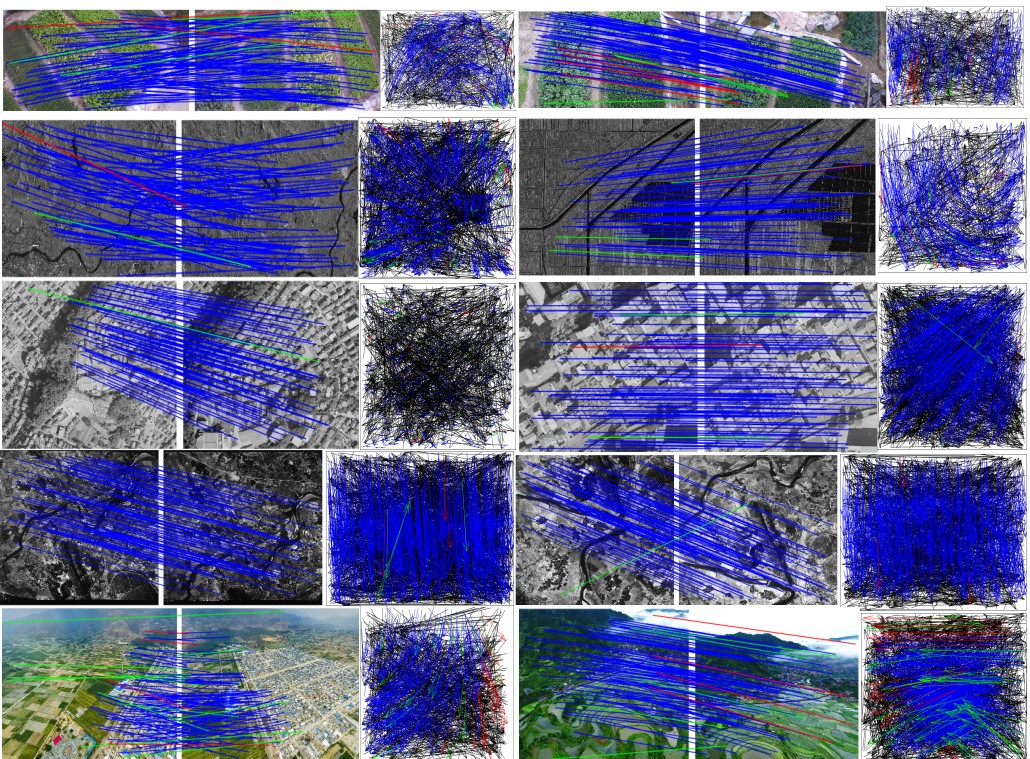

**Figure 6.** Feature-matching results of our LAF on 10 representative remote-sensing image pairs. ((**top**) to (**bottom**) and (**left**) to (**right**)) UAV1, UAV2, SAR1, SAR2, PAN1, PAN2, CIAP1, CIAP2, FE1, and FE2 (blue = true positive, black = true negative, green = false negative, and red = false positive). For each example, the graph on the left represents the intuitive result for the image pair, and the graph on the right represents the corresponding motion field.

*4.3. Results on Image Stitching*

Next, we compare our gray-level panorama with our previous work [34] in Figure 7. Zooming in on several local areas showed that our method eliminated ghostly and misaligned areas in the panorama.

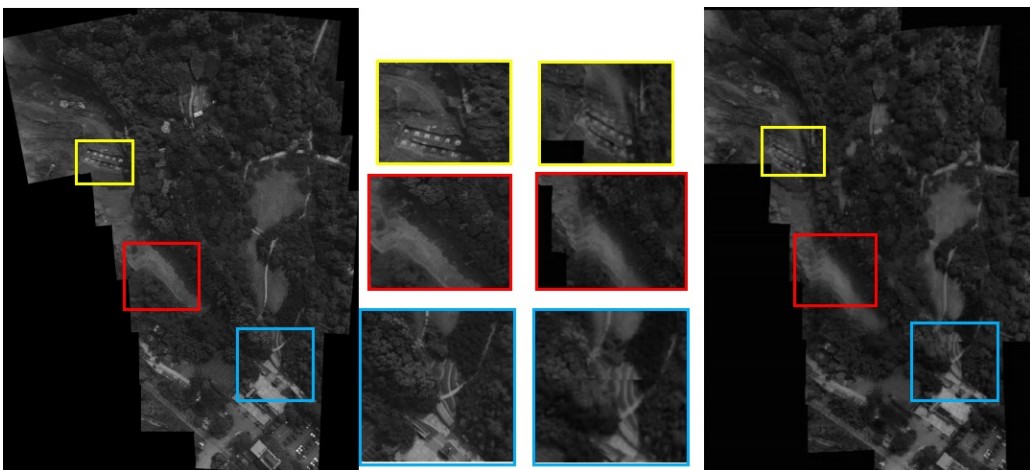

**Figure 7.** Comparison of the stitching results of the first set of hyperspectral images with the authors' previous work [34] (single band).

Then, we compared our proposed method with ANAP [25], NISwGSP [26] and ELA [27] in 3 groups of hyperspectral images, and each group containing 18 HSI images. We applied these algorithms to UAV-borne hyperspectral images and compared them with our method. These images contained large translation movements, resulting

in a large parallax. In Figure 8, ANAP [25], NISwGSP [26], and ELA [27] all showed low alignment accuracy with some representative areas represented by red boxes. In Figure 9, in the area shown in the red box, ANAP and NISwGSP had severe deformation, irregular amplification, or distortion at the edge. Although the ELA removed the ghost to some extent, one road was out of place. Figure 10 shows that the other algorithms also had varying degrees of ghosting and blurring in the house structure. ANAP relieved ghosts on some houses but reduced alignment accuracy. A large parallax in the overlapping region renders these algorithms unable to accurately align images, resulting in ghosts and fuzziness. Our method aligned the image more accurately, and eliminated the problems of deformation and distortion in the nonoverlapping region after stitching. Experiments show that our method could better solve these problems.

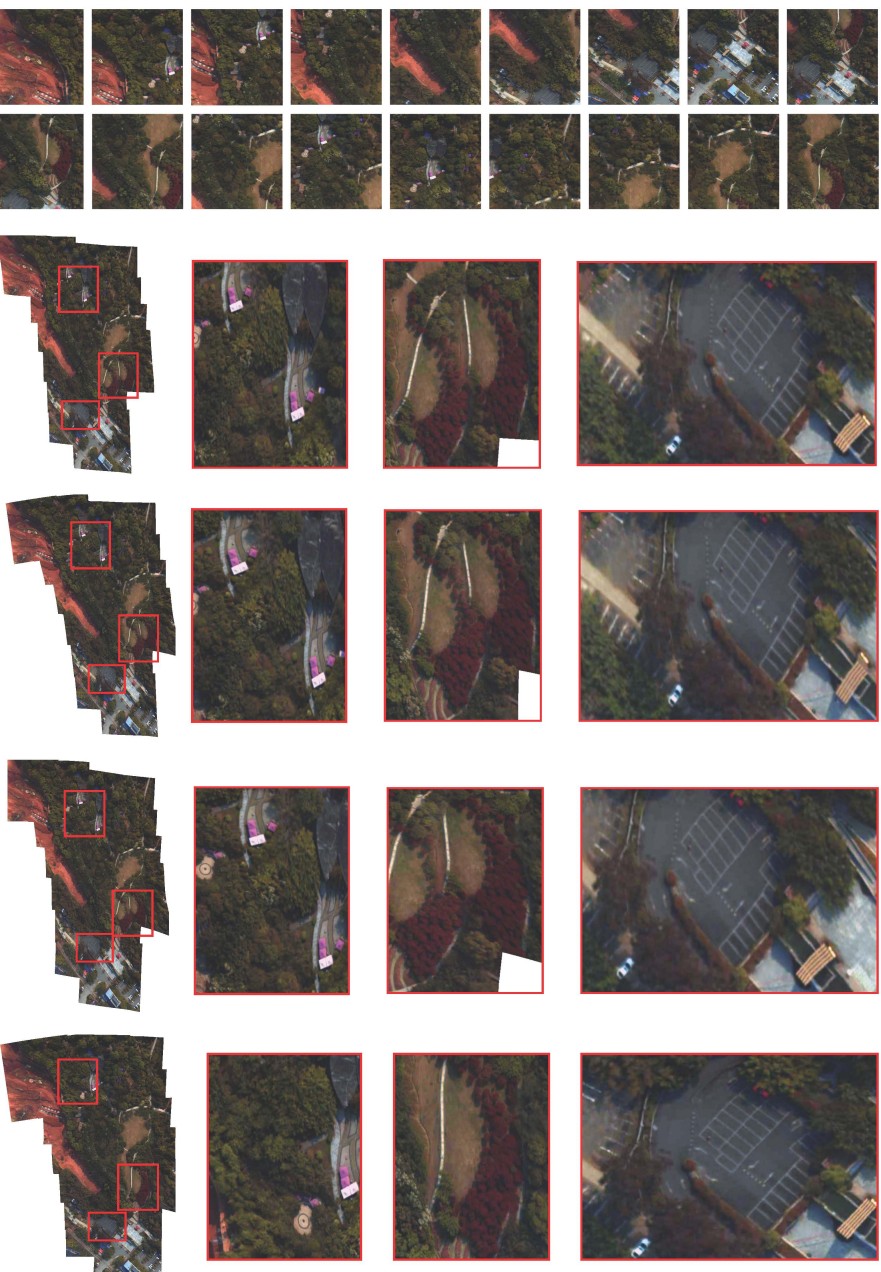

**Figure 8.** Comparison of the stitching results among ANAP [25], NISwGSP [26], ELA [27], and the proposed method (from (**top**) to (**bottom**)). At the top are the 18 original images to be stitched.

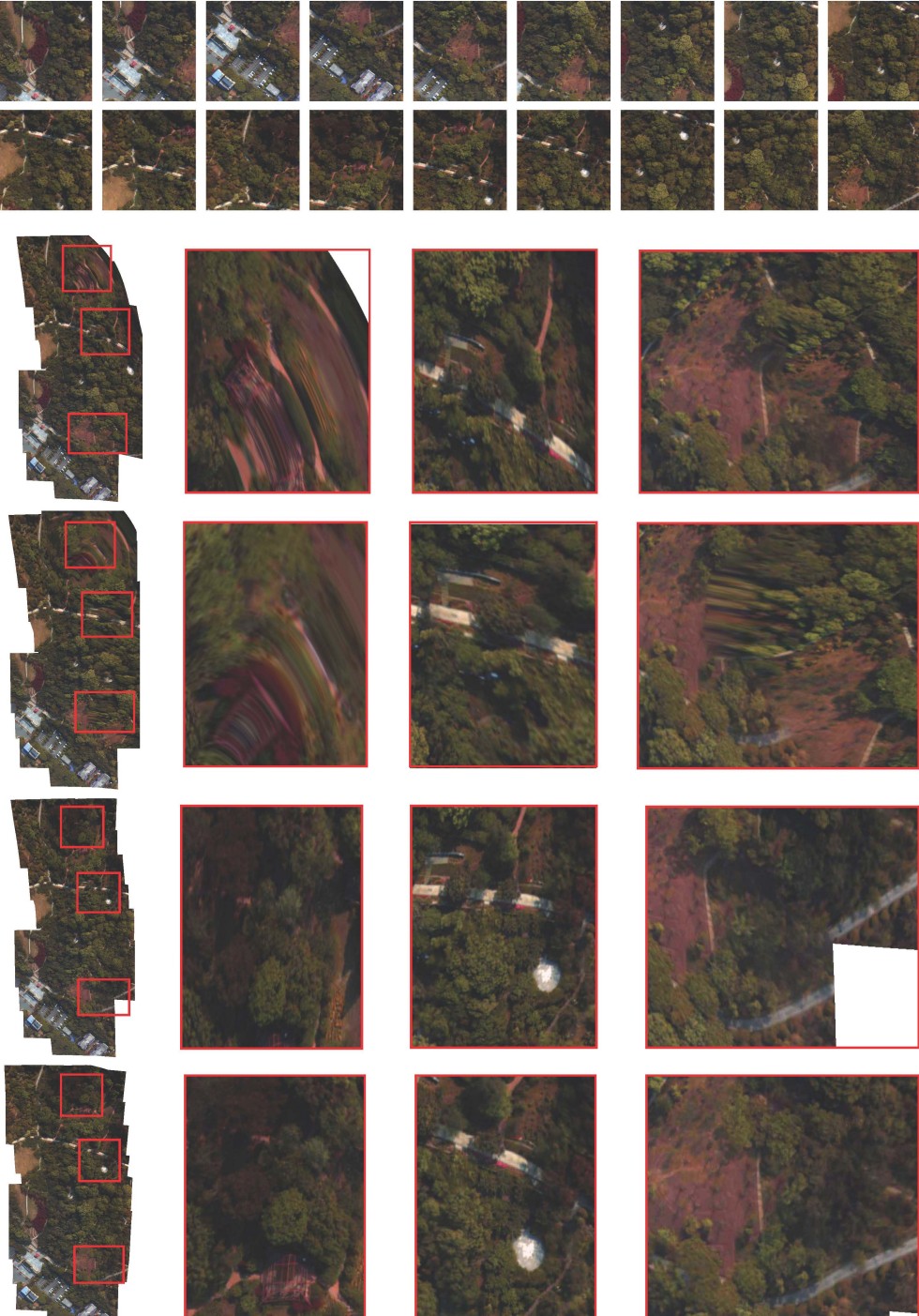

**Figure 9.** Comparison of the stitching results among ANAP [25], NISwGSP [26], ELA [27], and the proposed method (from (**top**) to (**bottom**)). At the top are the 18 original images to be stitched.

We used our method to stitch a total of 54 images in the HSI dataset and synthesize a pseudocolor image in Figure 11. Our method achieved a large-scale hyperspectral image stitching, while the other methods failed to produce the final results. This shows that our method generated a highly accurate panorama with satisfactory alignment quality.

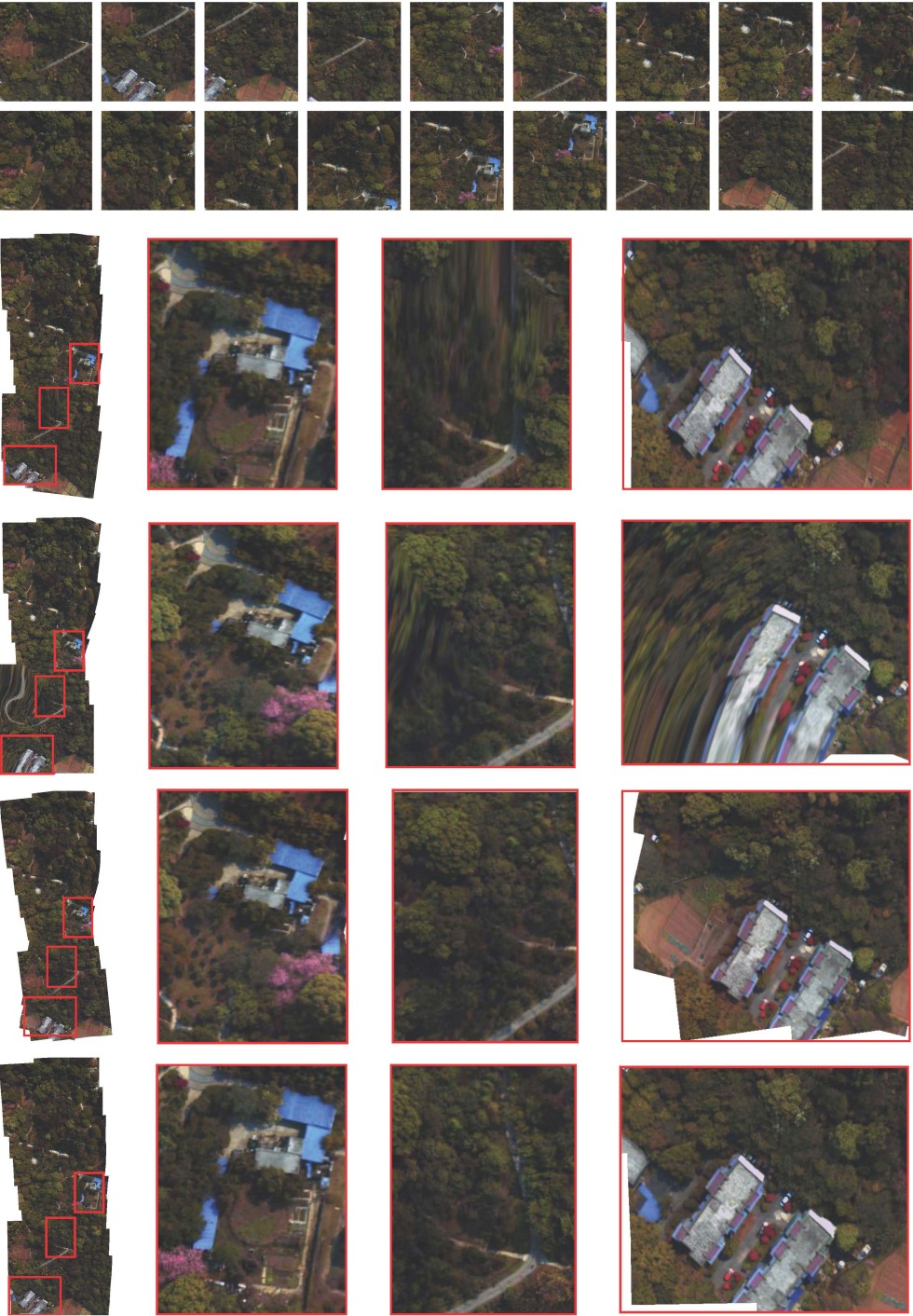

**Figure 10.** Comparison of the stitching results among ANAP [25], NISwGSP [26], ELA [27] and the proposed method (from (**top**) to (**bottom**)). At the top are the 18 original images to be stitched.

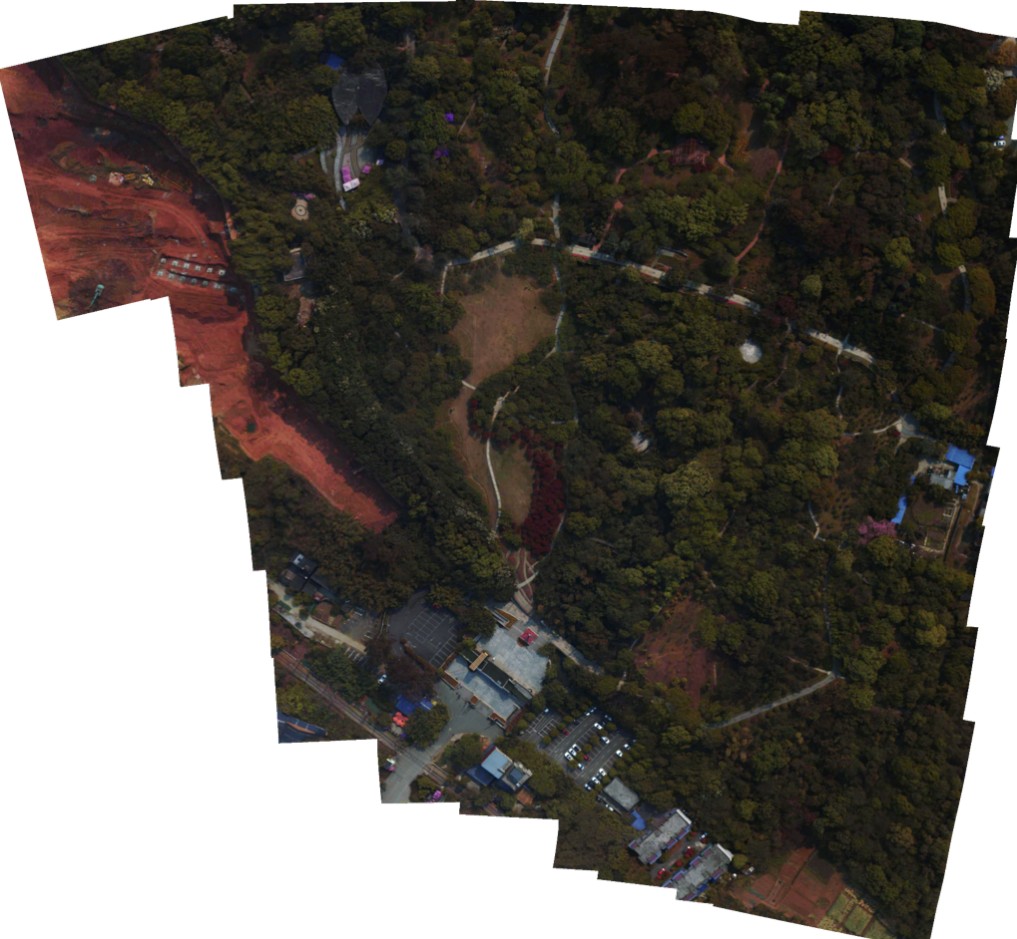

**Figure 11.** The final result of stitching 54 hyperspectral images in the HSI dataset using our method. We synthesized the pseudocolor image by selecting the 92nd (700.2 nm), 47th (547.6 nm), and 13th (436.5 nm) bands.

This section objectively evaluates image stitching quality. Since there was no reference image in the final panorama, we used some image-quality evaluation indices without a reference image, such as *variance*, *EOG*, and *DFT*. *variance* refers to the discrete degree of image pixel gray value relative to the mean value. If the variance is large, it indicates that the gray level in the image is scattered, and the image quality is high. *EOG* reflects the variation between gray scales. A large value represents multiple image layers and demonstrates the clarity of the image. *DFT* reflects the overall activity of the image space. The obtained experimental results are shown in Table 1:

**Table 1.** comparison of the evaluation indicators among four algorithms.

| Algorithm | *Variance* | *EOG* | *DFT* |
|:---:|:---:|:---:|:---:|
| ANAP | $3.172 \times 10^6$ | $2.577 \times 10^8$ | $3.194 \times 10^8$ |
| NISwGSP | $3.204 \times 10^6$ | $2.597 \times 10^8$ | $3.193 \times 10^8$ |
| ELA | $3.152 \times 10^6$ | $2.573 \times 10^8$ | $2.877 \times 10^8$ |
| Our method | $3.742 \times 10^6$ | $2.996 \times 10^8$ | $4.754 \times 10^8$ |

The above experimental results show that all evaluation indices of the images obtained through the image transformation model in this paper were improved, and that the image quality obtained in this paper is better.

*4.4. Spectral Analysis*

The spectral analysis of images can realize the classification and recognition of ground objects, so a stitching task should not only focus on spatial information, but also analyze the spectrum. Ideally, the spectrum of the panorama should be consistent with that of the reference image in the overlapping region. The similarity of two spectral curves can be judged by calculating the spectral angle distance ($SAD$) [39]. In this section, hyperspectral image data HSI was used for the experiments. First, first image $I_1$ and the second image $I_2$ are taken as examples to find two pairs of specific ground objects (land and vegetation) and are recorded as $A$, $A'$ and $B$, $B'$, respectively.

Since image $I_1$ is a reference, the spectrum of the panorama should be close to that of image $I_1$. Compared with the previous algorithm, the spectrum obtained by the new algorithm was closer to image $I_1$ (Figure 12). According to the calculation, $SAD$ of $I_1$ and $I_2$ at this point were 0.0894 for point pair $A'$ and $A'$. The $SAD$ of the image after stitching with the previous algorithm of the author and the image of 1 at this point was 0.0308, and the $SAD$ obtained by the algorithm proposed in this paper was 0.0084. The $SAD$ of $B'$ and $B'$ was 0.0471. The $SAD$ of the image after stitching with the previous algorithm was 0.0212 at this point, and the $SAD$ obtained by the algorithm of this paper was 0.0104. Therefore, the spectral values of the image obtained by the algorithm in this paper were closer to the reference image when the spectral values of the same point pair of the original image were significantly different, ensuring the consistency of the spectral information.

The above experimental results show that all evaluation indices of the images obtained through the image transformation model in this paper were improved, and that the image quality obtained in this paper was better.

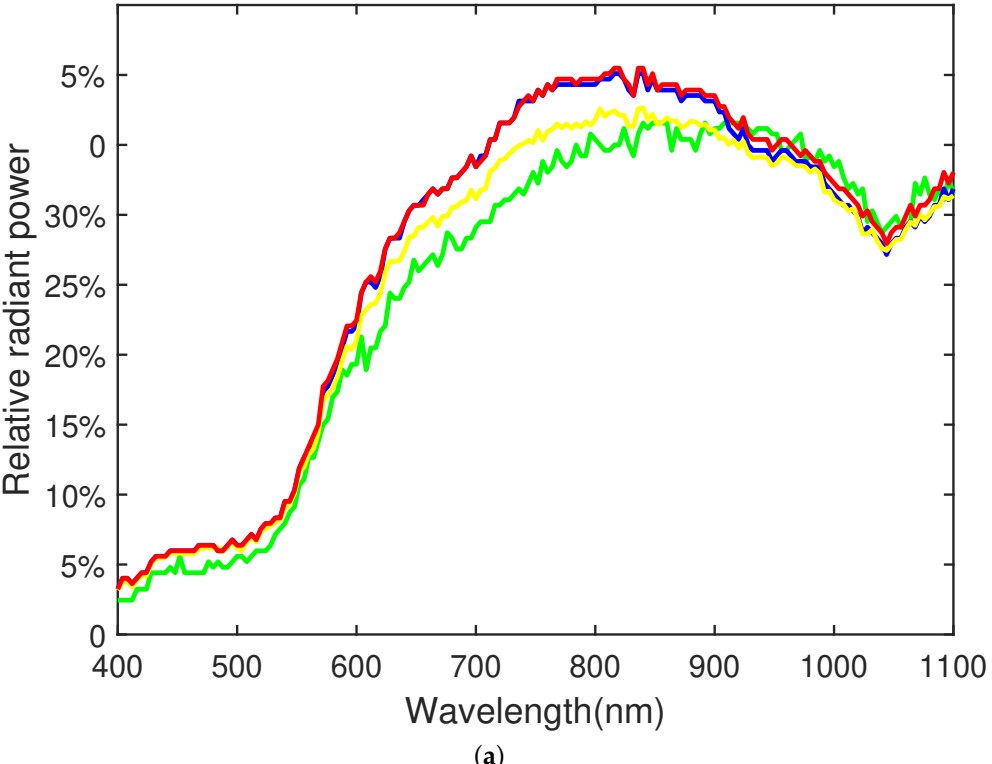

(**a**)

**Figure 12.** *Cont.*

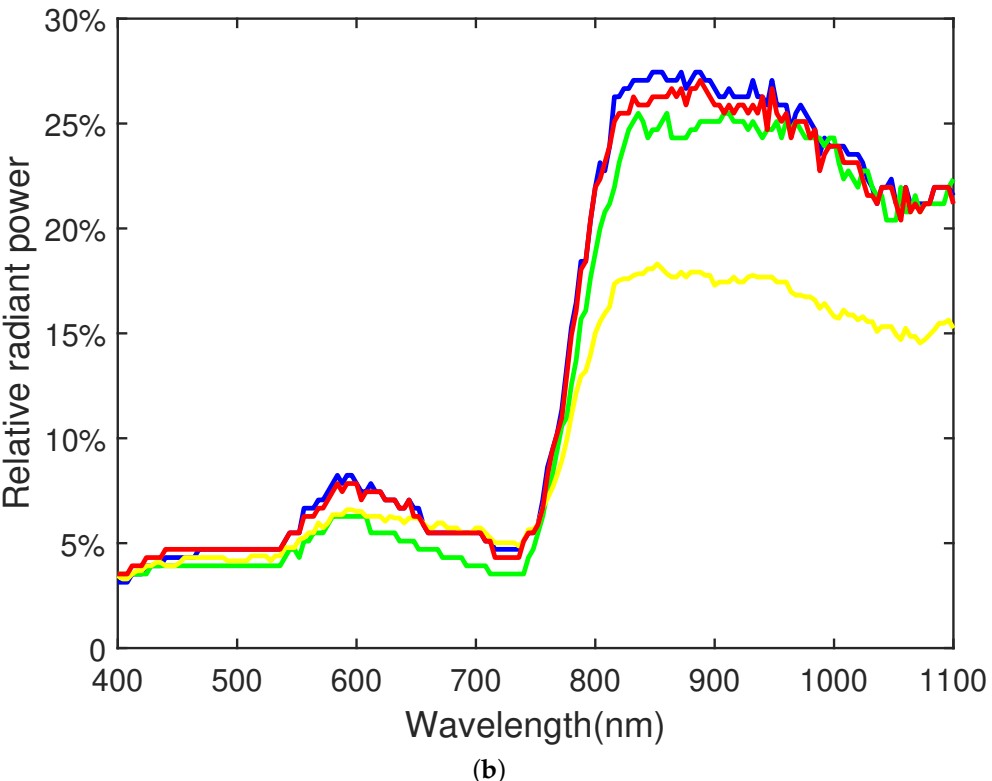

(**b**)

**Figure 12.** Spectral analysis. Blue and green curves are the spectrum of the two original images before stitching. Blue, spectrum of the reference image; yellow, the spectrum without spectral correction; red, the spectrum after spectral correction. (**a**) sift; (**b**) surf.

## 5. Conclusions

A novel image stitching algorithm for UAV-borne hyperspectral remote-sensing images is proposed in this article that focuses on improving accuracy and efficiency to render it suitable for multiple large-scale image-stitching tasks. Specifically, a deep local feature, SuperPoint, was integrated into the proposed feature detection. The LAF algorithm was proposed to establish accurate feature-point correspondences. The adaptive bundle adjustment was designed to solve accumulation errors and distortions. Lastly, we corrected the spectral information on the basis of covariance correspondences to avoid spectral distortion. The proposed approach achieved high accuracy and satisfactory quality on several challenging cases.

**Author Contributions:** Conceptualization, Y.Z. and J.H.; methodology, Y.Z. and X.J.; software, Y.Z. and Z.P.; writing—original draft preparation, Y.Z.; writing—review and editing, Y.Z., X.M. and J.H.; supervision, Y.M. and X.M. All authors have read and agreed to the published version of the manuscript.

**Funding:** This work was supported by the National Natural Science Foundation of China under grant no. 61903279.

**Conflicts of Interest:** the authors declare no conflict of interest.

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
