# Peer review of "Hyperspectral Panoramic Image Stitching Using Robust Matching and Adaptive Bundle Adjustment"

_remotesensing, doi:10.3390/rs14164038_

Round 1

Reviewer 1 Report

This manuscript focuses on hyperspectral panoramic image stitching by using mismatch removal approach and linear adaptive filtering. There are many issues or defects in it.
1.In line 29, what is "HSI"?
2.In line 156, x^(l,r)_k and x^(r.l)_k are used. But in the last second line of page 5, x^k_(l,r) and x^k_(r,l) are used.
3.In the last second line of page 5, S'  must be written in an equation.
4.In the last second line of page 5, what is the parameter "n"?
5.In the last second line of page 5, S' does not have the parameter "l", but x^i_l has the index "l".
6.In the last second line of page 5, m_i has the parameter "i", but x^k_(r,l)-x^k_(l,r) does not have the index "i".
7.In Eq. (3), what is the parameter "epsilon"?
8.What are the actual values of "epsilon" used in the experiments?
9.In the third line below Eq. (3), what is C_(j,k)?
10.In Eq. (4), what are the actual values of "beta" used in the experiments?
11.In Eq. (5), what is "lambda"?
12.In Eq. (9), what are the actual values of "tau" used in the experiments?
13.In line 186, what are g(x.y) and h(x.y)?
14.In line 186, there are two g(x.y). It is strange.
15.In line 191, what are g_i and h_i?
16.In line 191, there are two g_i. It is strange.
17.In Eq. (10). what are u_h and u_s?
18.In Eq. (10). what are the actual values of u_h and u_s used in the experiments?
19.In line 199, what are the actual values of "o" used in the experiments?
20.In Eq. (13), what does "ref" mean?
21.What are the differences between W_(j.k) in the third line below Eq. (3), W in Eq. (11), and W_i in Eq. (13)?
22."G" in Eq. (13) is different from "G" in the next line.
23.In Eq. (15), what do "theta" and "psi" mean?
24.In the line below line 220, {x_(l,r)l^k,x^k_(r.l)} is wrong.
25.In lines 260 and 261, five methods are used to compare for Fig. 5. But in Fig. 5, only four methods are used to compare. LPM [19] is lost.
26.In the caption of Fig. 4, "SS-SIFT [11]" is mentioned in the first line, but "SuperPoint [26]" is mentioned in the last line.
27.In line 271, "matching pairs" is mentioned for Fig. 6. But in Fig. 6, the j0 images in the third and sixth columns are not paired.
28.In line 275, [29] is used to compare for Fig. 7. But in the caption of Fig. 7, [44] is used to compare. Moreover, there is no [44].
29.In lines 278 and 279, ANAP [23], NISwGSP [24] and SPHP [33] are used to compare for Fig. 7. But in the caption of Fig. 8, ANAP [23], NISwGSP [24], ELA [25] are used to compare.
30.In the caption of Fig. 8, "from left to right" should be replaced by "from top to bottom".
31.For Fig. 8. correct images without stitching corresponding to the rectangular areas are needed to judge the stitching performances of the four methods. (This is requisite.)
32.In the caption of Fig. 9, "from left to right" should be replaced by "from top to bottom".
33.For Fig. 9. correct images without stitching corresponding to the rectangular areas are needed to judge the stitching performances of the four methods. (This is requisite.)
34.In the caption of Fig. 10, "from left to right" should be replaced by "from top to bottom".
35.For Fig. 10, correct images without stitching corresponding to the rectangular areas are needed to judge the stitching performances of the four methods. (This is requisite.)
In conclusion, major revision is necessary.

Author Response

We would like to thank the reviewers and the associate editor for their time and constructive comments, which enable us to greatly improve the quality of our manuscript. Next we provide point-by-point responses to the review comments. Please see our attachment.

Reviewer 2 Report

In the reviewed paper, the authors focused on the hyperspectral panoramic image stitching task. Image merging is a very needed operation when we made more than one image. Here, the authors propose using a reference band, a deep neural network. The idea is interesting, but some issues are missing:

-the abstract is very chaotic. Please, underline the proposition, how its works and its novelty

-is there any minimum number of points needed to stitch two images?

-your solution could be also used for side scan sonar images in remote sensing areas, see and analyze more such papers. For instance, similar idea was used in: side-scan sonar analysis using roi and deep neural networks

-comparison with state-of-art is needed

-comparison with classic solutions like to sift, surf, kaze, etc should be made.

-what are time/computational complexities?

-experimental section should be extended to time analysis.

-is it possible to publish your paper with code?

Author Response

(The authors gave the same response as above.)

Reviewer 3 Report

The author introduced a new method for hyperspectral imaging stretching using adaptive filter methods. I have few points to proceed further process:

1. The author used SURF, SIFT features for extracting features, these features has lot of drawbacks like angle mapping. So how the author used this techniques for stretching images in Hyperspectral. 

2. The author may include the advantages of Bundle adjustments methods.

3. What about stretching accuracy of our images.

4. In equation 3, the author may include, (⊗) point wise multiplication or tensor multiplications.

5. I think equation 7 and 8 is explanation is not mentioned in this paper.

6. which value you given for O constant in equation 12.

Author Response

(The authors gave the same response as above.)

Round 2

Reviewer 1 Report

This revised manuscript is ready to be published.

Reviewer 2 Report

All my comments have been addressed.